# Feasibility of a social protection linkage program for individuals at-risk for tuberculosis in Uganda

Grace Nanyunja[1], Jillian L. Kadota[2], Catherine Namale[1], Mollie Hudson[2], Talemwa Nalugwa[1], Stavia Turyahabwe[3], Adithya Cattamanchi[1,2,4], Achilles Katamba[1,5], Prosper Muhumuza[6], Priya B. Shete[1,2]*

1 Uganda Tuberculosis Implementation Research Consortium, Kampala, Uganda, 2 Division of Pulmonary and Critical Care Medicine and Center for Tuberculosis, University of California San Francisco, San Francisco, California, United States of America, 3 National Tuberculosis and Leprosy Program, Uganda Ministry of Health, Kampala, Uganda, 4 Division of Pulmonary Diseases and Critical Care Medicine, University of California Irvine, Irvine, California, United States of America, 5 Department of Medicine, Clinical Epidemiology & Biostatistics Unit, Makerere University College of Health Sciences, Kampala, Uganda, 6 Expanded Social Protection Programme, Ministry of Gender, Labor and Social Development, Kampala, Uganda

☯ These authors contributed equally to this work.
* Priya.shete@ucsf.edu

**Data Availability Statement:** Cleaned and deidentified data as well as relevant metadata and

## Abstract

Social protection interventions have the potential to accelerate progress towards global tuberculosis (TB) targets. We piloted a screening and linkage program at four community health centers (HC) to enroll adults seeking TB diagnostic evaluation services into existing government-supported social protection programs in Uganda. From May-December 2021, health center staff were asked to screen adults being evaluated for TB for eligibility for government-supported social protection programs, and to refer eligible people to a sub-county community development office (CDO) responsible for enrolling community members into government-supported social protection programs. Linkage was facilitated with a transportation reimbursement via mobile money and referral documentation confirming program eligibility. We assessed feasibility using programmatic data and conducted post-intervention surveys to understand experiences with the linkage program. Of 855 people undergoing TB evaluation, 655 (76%) adults met criteria for at least one government-supported social protection program. 25 (4%) of those were not interested in referral; the rest were referred to their local CDO. While 386 (61%) of the 630 participants reported to the CDO seeking social protection enrolment, only 122 (32%) of those were ultimately enrolled into a social protection scheme, representing only 19% (n = 655) of those eligible. In surveys conducted among 97 participants, 46 of the 60 (77%) people who reported that they sought enrollment at the CDO were not enrolled into a social protection program. Reasons provided for non-enrollment among these 46 participants were either unknown (n = 25, 54%) or due to operational challenges at the CDO including a lack of human resources or available groups to join in the social protection program (n = 20, 43%). 61 survey participants (63%) indicated that they would not have sought social protection enrollment without the referral program. Overall, we found that most adults seeking TB diagnostic evaluation are eligible for and interested in

data dictionary used in the analyses presented within this study are included in Supplement.

**Funding:** This work was supported by the UCSF-Gladstone Center for AIDS Research Mentored Science Award in HIV/AIDS Research (grant NIH P30 AI027763 Award A120163): PS; the Parker B. Francis Fellowship (grant P0523744): PS; and the Swedish Research Council (grant A134183): PS. The funders had no role in study design, data collection and analysis, decision to publish, or preparation of the manuscript.

**Competing interests:** The authors have declared that no competing interests exist.

obtaining government-supported social protection. We found facilitated linkage from HCs to CDOs offering social protection services to be feasible, however ultimate enrollment into programs was limited. Additional research is needed to identify strategies to improve access to existing social protection programs for eligible TB-affected individuals.

**Trial Registration:** Pan African Clinical Trials Registry (PACTR201906852160014).

## Introduction

Tuberculosis (TB) remains a major global health issue and among the leading causes of death worldwide. In 2021 alone, approximately 10.6 million people developed TB, causing almost 1.6 million deaths globally in the same year [1]. It is well-recognized that TB is as much a social disease as it is infectious, with those affected representing the poorest and most vulnerable individuals in society who disproportionately live in low and middle countries [1,2]. In addition to the financial vulnerabilities created by TB disease, evidence suggests that suffering from catastrophic costs due to TB, defined as costs surpassing 20% of annual household income, creates greater risk of adverse TB outcomes such as treatment failure, recurrence, and death, and perpetuates negative financial coping strategies such as dissaving, thus entrenching vulnerable individuals and households deeper into poverty and poor health [3–6].

Recognizing the interdependence between disease, social determinants of health, and socioeconomic development [5,7], both the United Nation's Sustainable Development Goals (SDGs) [8] and the World Health Organization's (WHO) 2015 End TB Strategy [9] encourage the implementation of innovative multisectoral approaches to address the intersection of social determinants of health and illnesses like TB [10,11]. Both highlight social protection and poverty alleviation as key strategies for addressing risk factors for acquiring TB and the economic consequences that follow its diagnosis [8,9]. Social protection is defined as social and economic policies and programs that target multiple dimensions of poverty among disadvantaged groups. They function by improving access to basic services and supporting livelihoods, thereby mitigating the consequences of financial shocks for those at risk of further impoverishment [12,13]. Examples include improving access to social services, assistance in the form of cash, food, or transportation vouchers, training for participation in income-generating activities, community administered savings groups, or training activities to encourage the development of human capital [13,14].

Beyond protecting vulnerable groups from deepening poverty due to ill health, there has been mounting interest in harnessing the benefits of social protection to achieve global TB elimination goals [15], particularly in Africa [14]. An emerging body of evidence demonstrates the utility and effectiveness of social protection for improving TB outcomes including increasing uptake of TB services [16], reducing risk of TB treatment default, and improving rates of in TB treatment success and cure [13]. Unfortunately, how best to link TB affected populations to social protection programs in high HIV/TB burden, low-income settings is unknown. The landscape of social protection in Uganda, exemplary of this context, includes a number of government-supported programs, but none that include a mechanism to enroll potentially eligible individuals affected by TB (so-called "TB-sensitive" programs), nor any social protection programs that target TB patients specifically (so-called "TB-specific" programs) (**S1 Table**) [17]. Limited evidence exists on how to design such programs in accordance with TB elimination strategies [9,18]. This gap presents a major operational and programmatic barrier for implementing social and economic supports for TB affected populations [19,20]. To address this gap, we implemented and evaluated the feasibility of a health center (HC)-based social protection linkage strategy for individuals presumed to have TB in Uganda.

## Methods

### Study setting and population

In partnership with the Ministry of Health (MoH) and the Ministry of Gender, Labour and Social Development (MoGLSD) of Uganda, we piloted a program to link eligible adults seeking TB diagnostic evaluation services from four community HCs with existing government-supported social protection programs in Uganda (May to December 2021). Participating HCs were classified as Level IV HCs, where TB diagnosis and treatment services are offered and managed by the National Tuberculosis and Leprosy Programme (NTLP) within the MoH. Participating community development offices (CDO, n = 10) served the same catchment area as participating HCs. CDOs are under the purview of the MoGLSD and are responsible for the local administration of government-supported social protection programs including disseminating information on existing social services locally, screening and enrolment for eligible individuals, and distributing benefits.

### Eligibility criteria and study population

The study population included all individuals ≥18 years of age being evaluated for TB at a participating HC, defined as having submitted a sputum sample to the laboratory for TB diagnostic evaluation. After submission of their sputum sample, health center staff asked adults three questions to screen them for program eligibility; these questions corresponded to enrollment criteria established by the seven major government-supported social protection programs included in our pilot program. This included verification that the individual: 1) lived in a participating subcounty served by one of the included CDOs; 2) was not a salaried government worker; and 3) was not already receiving benefits from a government-supported social protection program.

### Procedures

**Social protection program identification.** With key stakeholders including MoGLSD and NTLP, we landscaped the social development sector to identify active government-supported social protection programs being implemented across all regions where the pilot was taking place. We collected data on each social protection program according to target population, eligibility criteria, benefit scheme and structure, geographic catchment area, and duration of program implementation.

**Facilitated linkage process.** Procedures for social protection program facilitated linkage included steps for screening, eligibility determination, and provision of a transportation voucher and referral documentation for eligible adults to present to the local CDO. Research study staff introduced linkage procedures to each participating HC and subcounty CDO and conducted trainings with relevant personnel. At each HC, the village health team (VHT) worker responsible for screening HC clients for TB was identified to serve as the focal point for study activities. The VHT assessed all persons under evaluation for TB for eligibility criteria described above and informed each person whether they were likely eligible for a social protection program based on results of the screening. Those likely eligible for social protection program enrollment were asked about their interest in referral; those interested were given: 1) a referral slip with a unique identifier to present to the CDO officer with sociodemographic and eligibility information, and 2) instructions on how to access the respective subcounty CDO to seek social protection program enrollment. VHTs also informed all participants that they would receive a one-time transport voucher of 20,000 Ugandan Shillings (USH; ~$5.64USD) after they presented to the CDO. At the CDO, the development officer received the

participants with the referral slip bearing their unique identifier, registered them in a study screening and enrollment logbook, and then followed routine procedures for enrolling community members into government-supported social protection schemes including providing information on available schemes and prerequisites for enrolment. At the CDO, participants found eligible for one or more schemes were enrolled as per the discretion of the development officer and according to their standard operating procedures.

### Data collection

**Social protection referral data collection.**   Process metrics were abstracted from logbooks at each HC and CDO to assess implementation and enrollment outcomes (see **Outcomes**). Focal persons at each HC and CDO was identified and trained using a camera-enabled smartphone to take photos of the HC and CDO logbooks, respectively. They would then upload the photos to a central secure server using the National Institutes of Health (NIH)-recommended Research Electronic Data Capture (REDCap) software mobile application [21].

**TB clinical data collection.**   Full data abstraction procedures are standardized for TB diagnostic evaluation studies within the Uganda Tuberculosis Implementation Research Consortium (U-TIRC), and are described elsewhere [22]. In brief, data were collected at all HCs from NTLP Presumptive TB registers, Laboratory and Treatment registers, and Xpert laboratory requisition forms. These data included key patient demographic and clinical (including HIV-related) information in addition to TB outcomes. Data quality assurance is as described previously [22]. Research staff matched social protection referral data to TB register data using participant identifiers such as name and referral dates to create a linked study database which was subsequently de-identified.

**Survey data collection.**   The VHT also assessed those found to be eligible for government-supported social protection programs for interest in survey participation. We randomly selected a subset of those interested for survey participation, stratified by HC and sex. We collected socioeconomic information as well as perceptions and experiences with linkage to the CDO for social protection enrollment. Surveys were conducted one month following HC referral.

### Outcomes

Study outcomes focused on implementation outcomes reflected in the HC screening and linkage process. These outcomes included numbers and proportions of: 1) people with presumed TB screened for eligibility for study and social protection linkage criteria at the HC (screened), 2) those eligible for linkage among total screened (eligible), 3) those referred to the CDO among those eligible (referred), 4) those who presented to the CDO after referral (linked), 5) those sent the 20,000 USH transport reimbursement, and 6) those ultimately enrolled into a social protection scheme (enrolled). Taken together, these implementation outcomes were used to interpret the overall feasibility of the linkage program. In addition, we assessed technical feasibility of HC screening for program eligibility by comparing those who were found eligible at the HC to those found eligible at the CDO. The proportion of those who were eligible for a program, interested in being referred and ultimately accepted a referral to a CDO was used to assess linkage program acceptability. Among those who presented to the CDO we ascertained time from referral to presentation for social protection enrollment. We additionally included TB care outcomes related to TB screening, diagnosis, and treatment.

### Statistical analysis

We assessed patient characteristics, implementation of the pilot linkage process, TB register data, and survey results using numbers and proportions for categorical data and either median

and interquartile ranges (IQR) or means with standard deviations (SD) for continuous variables. All analyses were conducted using Stata version 16 (StataCorp, TX USA).

The study protocol was approved by the institutional review boards (IRB) of the University of California San Francisco, San Francisco, CA, USA, Makerere University School of Public Health, Kampala, Uganda and by the Uganda National Council for Science and Technology, Kampala, Uganda, and included a waiver of written informed consent for social protection program referral and access to routinely collected demographic and clinical information among all participants. Verbal consent was obtained for participation in surveys by an experienced study staff member trained in human subjects research. Study staff members called participants and used an IRB-approved verbal consent form script to administer the consent form. Participants' responses were audio recorded using REDCap.

## Results

### Social protection program characteristics

We identified seven major government-supported social protection programs being implemented across all regions where the pilot was taking place. Briefly, these programs included: Youth Livelihood Program, Uganda Women Entrepreneurship Program, Emyooga, Parish Development Model, Social Assistance Grant for Empowerment, People with Disabilities, and Operation Wealth Creation. A more detailed description of the target population and intended function and structure of benefits and geographic coverage are summarized in **S1 Table**. All programs provide benefits at either the subcounty or parish level, which is the most granular administrative level in a sub-county. The majority of the included social protection programs were group-based, in which benefits are distributed not individually, but to a group of eligible individuals participating in the same enterprise. Most social protection programs provided benefits as loan requiring repayment with interest. Two programs distributed benefits as grants, which do not require repayment, and one program distributed agricultural inputs to land-owning beneficiaries.

### Feasibility of the pilot facilitated linkage process

From May to December 2021, we screened 855 persons undergoing TB diagnostic evaluation at participating HCs. Among the 855 screened, we found 655 (76%) adults to be eligible per government-supported social protection program criteria. Reasons participants were found to be ineligible included those living in a subcounty outside of the jurisdiction of the CDO (n = 134/855; 16% of all screened), those not meeting age requirements (n = 38/855; 4% of all screened), those who were government workers on payroll (n = 14/855; 2% of all screened), and those already enrolled in social protection schemes (n = 14/855; 2% of all screened). 25 (4%) of those eligible were not interested in referral and declined to participate. The remaining 630 (96%) who were interested in referral were informed they were likely eligible for a social protection program based on results of the screening. All were referred to the CDO for further assessment. Sociodemographic and clinical information on participants referred for social protection enrollment is shown in **Table 1**. 386 (61%) of those referred were successfully linked to the sub-county CDOs and were ultimately evaluated for social protection program enrollment with a median time to linkage of 1 day (IQR: 0–4). The development officer at the CDO found 382 of the 386 (99%) participants referred for social protection enrollment to be eligible for one or more government-supported social protection program. Of the 386 total participants linked to the CDO, 362 (94%) received the transport refund. 122 (32%) of those who were linked and evaluated at the CDO were ultimately enrolled into a social protection scheme,

**Table 1. Demographic and clinical characteristics of the study population across all participating health centers.**

| Characteristic | N (%)(N = 630) |
|---|---|
| **Age**, *Median (IQR)* | 35 (26–48) |
| **Sex** | |
| Female | 372 (59.0) |
| Male | 258 (41.0) |
| **HIV status** | |
| Positive | 170 (27.0) |
| Negative | 441 (70.0) |
| Unknown | 19 (3.0) |
| **Micro-bacteriologically confirmed TB (MTB)** | |
| Yes | 36 (5.7) |
| No | 594 (94.3) |
| **Treatment outcome[1]** | |
| Not treated | 600 (95.2) |
| Completed | 15 (2.4) |
| Cured | 10 (1.6) |
| Transferred out | 1 (0.1) |
| Lost to follow up | 1 (0.1) |
| Died | 3 (0.5) |

CDO = community development office; IQR = interquartile range; TB = tuberculosis.

representing only 19% of those eligible (**Fig 1**). There was substantial variation in implementation outcomes by health center (**S2 Table**).

Definitions for treatment outcome categories are as follows: 1) Not treated: A patient with no evidence of initiation of treatment; 2) Completed: A patient who completed treatment as

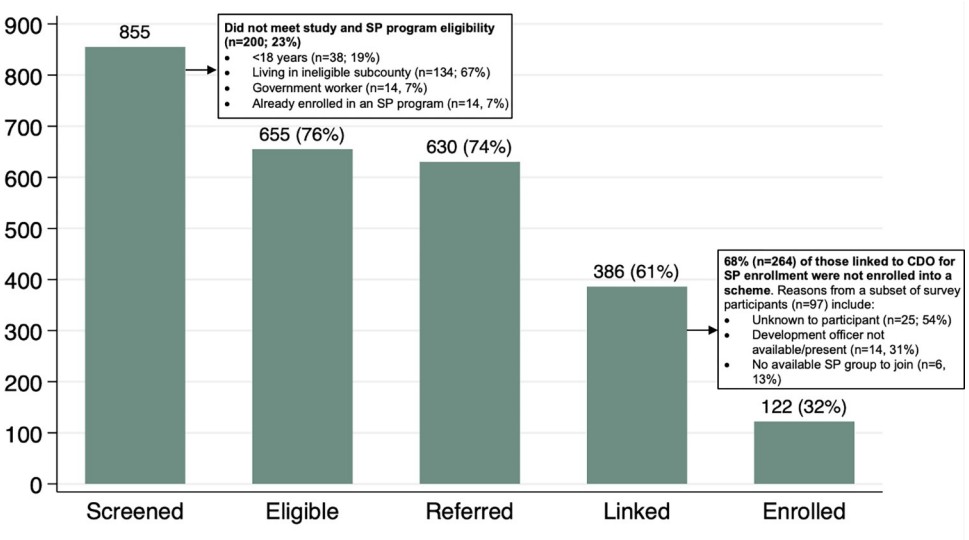

**Fig 1. Linkage feasibility based on cascade of screening, referral, and enrolment into a social protection program.**
CDO = community development office; SP = social protection. Each bar represents a necessary step in the linkage program from screening for social protection enrolment, identification of those eligible, facilitated referral for those interested in participating, linkage to the CDO, and enrolment in an available program.

recommended by the national policy whose outcome does not meet the definition for cure or treatment failure; 3) Cured: A pulmonary TB patient with bacteriologically confirmed TB at the beginning of treatment who completed treatment as recommended by the national policy with evidence of bacteriological response and no evidence of failure; 4) Transferred out: A patient who initiated treatment but transferred out to an alternate before completion, and for whom no data is available on treatment outcome; 5) Lost to follow-up: A patient who did not start treatment or whose treatment was interrupted ≥2 consecutive months; 6) Died: A patient who died before starting treatment or during the course of treatment [23].

## Survey results

Ninety-seven participants (median age was 38 years (IQR: 26–48); 57 (59%) were female) responded to our survey. Over half (n = 49; 51%) of the participants were multidimensionally impoverished or severely multidimensionally impoverished based on the Multidimensional Poverty Index (MPI) (22). Dissaving due to the economic impact of TB was common; 74% (n = 72) used personal or household savings, 51% (n = 50) reduced household food intake, 41% (n = 40) sold an asset, and 40% (n = 39) took out a loan to cover costs associated with their current episode of illness (**Table 2**). Unmet social needs were also commonly reported; 70% (n = 68) of participants had food insecurity, 39% (n = 38) lacked reliable transport to the HC, and 80 participants (82%) indicated that it was difficult to pay or find basic needs such as food, housing, and medical care (**Table 3**).

**Table 2. Household and individual characteristics of survey participants.**

|  | N (%) |
|---|---|
| **Total surveys** | 97 |
| **Age**, *Median (IQR)* | 38 (26–48) |
| **Sex** |  |
| Female | 57 (58.8) |
| Male | 40 (41.2) |
| **Multidimensional Poverty Index** |  |
| Not poor/not vulnerable to poverty | 17 (17.5) |
| Vulnerable to poverty | 31 (32.0) |
| Impoverished | 34 (35.1) |
| Severely impoverished | 15 (15.5) |
| **Participant is head of household** | 62 (63.9) |
| **Work status** |  |
| Currently working | 78 (80.4) |
| Not currently working | 19 (19.6) |
| **Education level** |  |
| None | 6 (6.2) |
| Primary level | 62 (63.9) |
| Secondary level | 25 (25.8) |
| Tertiary level | 4 (4.1) |
| **Household size**, *Mean (SD)* | 5.8 (2.9) |
| **Bank account status** |  |
| Has a functional bank account | 19 (19.6) |
| Does not have a functional bank account | 78 (80.4) |

IQR = interquartile range; SD = standard deviation.

**Table 3. Negative financial consequences and unmet social needs among individuals presumed to have TB (n = 97).**

| Economic impact of TB | N (%) |
|---|---|
| *Dissaving* | |
| Used personal or household savings | 72 (74.2) |
| Reduced household food intake | 50 (51.6) |
| Sold an asset | 40 (41.2) |
| Took out a loan | 39 (40.2) |
| Took a child out of school | 7 (7.2) |
| *Unmet social needs* | |
| **In past 1 month:** | |
| Worried food would run out (sometimes/often) | 68 (70.1) |
| Lack of reliable transportation kept from: health facilities, work, market, or from getting things needed for daily living | 38 (39.2) |
| Been to a place where free food is distributed | 14 (14.4) |
| *Current living situation* | |
| Steady place to live | 57 (58.8) |
| Steady place to live, but worried will lose it | 37 (38.1) |
| No steady place to live | 2 (2.1) |
| Missing living situation | 1 (1.0) |
| *Difficulty in paying/finding basic food/housing/medical care* | |
| Somewhat difficult | 69 (71.1) |
| Not difficult at all | 17 (17.5) |
| Very difficult | 11 (11.3) |

TB = tuberculosis. Participants in the pilot social protection linkage program randomly selected to participant in a survey described persistent social and financial challenges during their illness.

We compared self-reported data regarding social protection program referral, linkage and enrollment between HC and CDO logbooks. All survey participants (n = 97) were found eligible for CDO referral, 60 (65%) presented to the CDO to register for a program, but ultimately only 14 (14%) were enrolled into a scheme. Four survey participants did not realize they had been found eligible and were referred by health center staff to the CDO for social protection enrollment. Eight survey participants whose logbook data reflected social protection program enrollment self-reported that they were not enrolled, while 9 survey participants self-reported that they were enrolled into a program when logbook data indicated that they were not. Among the 60 survey participants who presented to the CDO to register, 46 were not enrolled into a program (77%), Most of these participants (n = 25; 54%) reported that they did not know the reason for why they were not enrolled (**Fig 1**). 63% (n = 61) of all survey participants indicated that they would have not sought out a social protection program without this linkage program. In an exploratory analysis of only those individuals who participated in our survey, we found that only age was associated with enrollment into a program; on average, those enrolled were younger compared to those who were not (28.9 years versus 40.0 years, respectively).

## Discussion

Innovative, multisectoral approaches are urgently needed in order to reach the ambitious global public health goal of ending TB. We sought to evaluate the acceptability and feasibility of a program in which we facilitated the linkage of people with presumed TB from community

health centers to local community development offices for enrollment into government-supported social protection programs. Our results suggest this linkage program was highly acceptable, with 96% of those eligible interested in the referral program, as well as technically feasible, with 99% of participants linked from the HC subsequently confirmed to be eligible for one or more government-supported social protection program. Our program delivered transportation vouchers with high fidelity; 94% of those who sought social protection enrolment at the CDO successfully received their mobile money. Our results also suggest that this social protection linkage program addressed an important need: 63% of surveyed participants felt that the program facilitated access to social protection benefits that they might otherwise not have sought out. Despite feasibility and acceptability of this health system-based linkage process, ultimate enrollment into programs was limited. Among the total eligible persons linked to CDOs, less than 20% of those who were eligible to receive social protection benefits were enrolled into a program.

Our study demonstrates individuals with presumed TB are exceedingly socioeconomically vulnerable, with high levels of impoverishment and unmet social needs. Over half of our participants were classified as either impoverished or severely impoverished based on a validated quantitative measure, the MPI. Surveys revealed many participants engaged in negative financial coping strategies, called dissaving, in order manage the costs of TB associated care seeking and symptoms, and that a vast majority faced unmet social needs including access to basic needs and services including food, housing, and medical care. These results are in keeping with the large literature citing the social and economic consequences of TB [3,6,24]. Unfortunately, while the majority of individuals being evaluated for TB related symptoms were eligible for social protection programs, less than 20% were enrolled, representing a major gap in social protection service delivery for this vulnerable population.

Several implementation challenges likely contributed to the substantial attrition of participants from the time of referral at the health center to social protection program enrolment. Distance between the HC and CDO and/or associated time and opportunity costs for additional visits to a government office may not have been overcome by our transportation vouchers for all participants. Among those that reached the CDO but were not enrolled in a social protection program, attrition was largely due to unpredictable closures at the CDO and a lack of available development staff. Even after successfully presenting to a development office for program enrollment, our survey results revealed that 20% of eligible participants were ultimately not enrolled into any program due to systems barriers, including strict limitations to enrollment based on the terms and structure of government-supported programs (e.g., there was no available group to join, or that the officer at the CDO was too busy to enroll them into a program). In most Ugandan social development programs, eligible clients are required to either form or join groups of 7 or more other eligible clients prior to being enrolled. Although requirement for group formation is likely intended to mitigate financial risk for the program and support community building, it presented a formidable barrier to enrollment for our study participants who are often left to navigate these complex terms on their own. Finally, we found evidence of knowledge and communication barriers among participants. Over half of survey participants indicated that they were not told the reason for why they were not enrolled into a social protection program. Other discrepancies noted in the survey, including mismatches between self-reported program enrollment versus actual enrollment based on logbook data, may reflect a breakdown in communication and lack of understanding among community members of the process of receiving social protection benefits. These results also highlight the importance of optimizing performance for social protection programs as a pre-requisite for successful implementation and scale up of sustainable multi-sectoral linkage programs that address the social and economic determinants of TB.

As with most pilot programs, our approach has several potential limitations. First, we did not include informal, community-based social protection programs that are common in Uganda, including saving circles or loans from faith-based organizations, nor did we include large social protection programs administered and funded by multilateral or bilateral implementing partners in our linkage program. We chose not to include these programs as they were not government funded and thus had less likelihood for sustainable scale-up. Our landscaping found only three available programs classified as "social protection programs" supported by the Government of Uganda, SAGE, which enrolls only the very aged, People with Disabilities, which enrolls only individuals with recognized physical disability, and the Parish Development Model, which was not yet fully operational at the time of this study. To overcome the limitation of too few programs, we also included more traditional social development programs. Future research may consider the impact that access to such programs may have on TB outcomes. Additionally, we observed high levels of heterogeneity in screening, eligibility, and enrollment across participating HCs but were unable to explain the reason behind these differences with our study data alone. Our exploratory analysis demonstrated that those enrolled were younger than those that were not; however as that analysis was of a small number of participants the results may not be generalizable. Additional research including interviews with health center staff and CDO officers will be necessary to gain a better understanding for the differences we saw in program eligibility and enrollment across health centers, as well as the supply and demand mismatch we observed in access to social protection programs for eligible TB-affected individuals overall.

The main strength of our study stems from our early partnership and collaboration with both the NTLP/Ministry of Health and the Expanded Social Protection Programme within the MoGLSD. Engaging a key governmental partner outside of the health sector with the mandate for provision of social protection programs enabled our study to co-design our implementation strategy, collect essential process metrics, and leverage ongoing programs in our approach. Our focus on implementation outcomes provides a simple and systematic approach to measure social protection program coverage and attrition in services for eligible TB-affected individuals which may be replicated in other similar settings. Finally, our use of explanatory surveys provides some crucial insights into key gaps in the linkage process, while also highlighting barriers and facilitators that will support the development of improved linkage strategies and metrics to foster multisectoral accountability, an approach highlighted as critical for reaching END TB targets [20].

While all government-supported social protection programs that were included in this study could be considered TB-sensitive, in that although not directly targeting TB-affected patients their effects could have downstream impacts on TB-related outcomes and mitigate the financial consequences of disease [19], the challenges with even facilitated linkage to such programs suggests that an improved strategy for reaching vulnerable TB affected individuals is required. Like other high-burden and low-income countries, no government-supported social protection programs are available to Ugandans with TB specifically on the basis of their TB status. Given the recent growing interest in the use of social protection and other multisectoral approaches to improve outcomes for TB-affected households in Uganda [20] and in other similar contexts [13,25], policy makers and stakeholders should consider collaborating to improve program performance for those eligible and/or expanding the types of social development and social protection programs available for individuals and households affected by TB.

## Conclusions

Individuals at risk for TB in this high-burden, low-income setting are extremely socioeconomically vulnerable and, unfortunately, not covered by government-supported social protection

programs despite eligibility and interest. A health center based social protection linkage program for individuals presumed to have TB, in partnership with community development offices, while acceptable and technically feasible, has marginal impact due to implementation barriers to enrollment within the development sector. Additional research is needed to identify optimal ways of enhancing access to essential social protections for this vulnerable population.

## Supporting information

**S1 Table. Characteristics and dates of implementation of government-supported social protection programs included in the linkage program.** Characteristics reflect data collected in May-December 2021.
(DOCX)

**S2 Table. Screening and referral metrics among linked participants by health center.** CDO = community development office; HC = health center; IQR = interquartile range.
(DOCX)

**S1 Data. A dataset including participant demographic, clinical, and social protection facilitated linkage process metric data.**
(CSV)

**S2 Data. A data dictionary describing the variables included in the S1 Data dataset.**
(XLSX)

## Acknowledgments

The authors are grateful to the administration, staff, and patients at the participating health centers for their time and participation, the staff and administration for the participating community development offices, the research administration of the Uganda TB Implementation Research Consortium, the Expanded Social Protection Programme within the Ministry of Labour, Gender, and Social Development, and the National Tuberculosis and Leprosy Program.

## Author Contributions

**Conceptualization:** Grace Nanyunja, Jillian L. Kadota, Mollie Hudson, Achilles Katamba, Priya B. Shete.

**Data curation:** Grace Nanyunja, Catherine Namale, Talemwa Nalugwa.

**Formal analysis:** Jillian L. Kadota.

**Funding acquisition:** Adithya Cattamanchi, Achilles Katamba, Priya B. Shete.

**Investigation:** Stavia Turyahabwe, Achilles Katamba, Prosper Muhumuza, Priya B. Shete.

**Methodology:** Stavia Turyahabwe, Adithya Cattamanchi, Prosper Muhumuza, Priya B. Shete.

**Project administration:** Grace Nanyunja, Jillian L. Kadota, Talemwa Nalugwa.

**Supervision:** Achilles Katamba, Prosper Muhumuza, Priya B. Shete.

**Writing – original draft:** Grace Nanyunja, Jillian L. Kadota, Priya B. Shete.

**Writing – review & editing:** Adithya Cattamanchi, Achilles Katamba.

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
