## [Decision Letter · Decision Letter 0]

9 Aug 2023

PGPH-D-23-01088

Feasibility of a social protection linkage program for individuals at-risk for tuberculosis in Uganda

Dear Dr. Shete,

Thank you for submitting your manuscript to PLOS Global Public Health. After careful consideration, we feel that it has merit but does not fully meet PLOS Global Public Health’s publication criteria as it currently stands. Therefore, we invite you to submit a revised version of the manuscript that addresses the points raised during the review process.

We look forward to receiving your revised manuscript.

Kind regards,

Lei Gao

Academic Editor

Journal Requirements:

1. In the ethics statement in the Methods, you have specified that verbal consent was obtained. Please provide additional details regarding how this consent was documented and witnessed, and state whether this was approved by the IRB.

Additional Editor Comments (if provided):

Please submit a revision considering all the comments from the reviewers, and point-to-point responses needed.

Reviewers' comments:

Reviewer's Responses to Questions

**Comments to the Author**

1. Does this manuscript meet PLOS Global Public Health’s publication criteria? Is the manuscript technically sound, and do the data support the conclusions? The manuscript must describe methodologically and ethically rigorous research with conclusions that are appropriately drawn based on the data presented.

Reviewer #1: Yes

Reviewer #2: Partly

2. Has the statistical analysis been performed appropriately and rigorously?

Reviewer #1: No

Reviewer #2: No

3. Have the authors made all data underlying the findings in their manuscript fully available (please refer to the Data Availability Statement at the start of the manuscript PDF file)?

Reviewer #1: No

Reviewer #2: No

4. Is the manuscript presented in an intelligible fashion and written in standard English?

Reviewer #1: Yes

Reviewer #2: Yes

5. Review Comments to the Author

Reviewer #1: No major statistical analysis is performed for this submission, it consists of summary data only.

Data for the analysis is available on request to the authors.

Lines 137-139: Were people in other formal employment eligible for a social protection program?

Lines 155-157: Was screening conducted by VHT based on the inclusion criteria described on line 137-139?

Table 1: Please include the denominator for results shown.

Lines 255-259: This sentence should appear earlier in the results, and in this section before presenting the survey results, the authors should include data and reasons for non-enrolment for the 264 who were not enrolled.

Lines 278 - 280: Consider rephrasing this sentence for clarity. It's unclear where the denominator 14 arises from, is this from the 97 participants who were surveyed? Also unclear what the total number out of 97 were actually enrolled in the program.

Lines 281: What is the denominator for 54% based on?

Lines 311 - 316: Appears to be a repetition of results and should be in the results section.

Lines 328-330: It's unclear what the 20% is based on?

Lines 336 - 339: Appears to be new data that should be in the results section.

Discussion: The authors state their data did not provide reasons for the mismatch in supply and demand observed in addess to social protection programs for those who were referred. Was any attempt made by investigators beyond their data collection to find out why a majority of patients referred for social program linkage were not enrolled or to provide a reason to those who were surveyed and reported not knowing why they did not get enrolled?

References:

Line 448: Please provide a date when the online resouce was accessed.

Line 504: Please provide a date when the online resource was accessed.

Reviewer #2: The authors addressed a very important problem that patients with tuberculosis and their families face related not only with the disease but related with social aspects including economic situation.

There are some recommendations to improve this work and to make it suitable to be published. In general although english is good, there are some typos and grammatical issues, that can be reviewed to improve the quality of the text.

In methods, it is not clear how was estimated the sample size for the survey, or what was the power of this sample. There are some analysis that could be proposed at least bivariate analysis even multivariable analysis, to better understand the relationship between some characteristics of the participants and actual access to social programs. The authors describe a survey with three questions to identify who is eligible to receive support for social programas, however it is not clear how they choose these questions, or if this questionnaire is validated.

In results section, the tables 1 and 2 can be easier to understand if row titles do not repeat information that is included in first column, like n(%). Also, it would be desirable to include every category of the variables, because this allows to evidence that there was no missing data. Also for ordinal qualitative variables, it is recommended to present their categories in the corresponding order. Since a small number of programs were identified, they can be listed within the text and then refer to the Table for more information. It would be useful to have a cascade graphic to show how many patients were screened at the beginning and how many got the support. The table S2 has very important information that is recommended to include in results section and not like a supplementary table. Image quality of figure 1 could be improved to better show the results.

In discussion section there are some data presented as result of the present study but those data where not described in the results section. Also it is recommended do not reference tables in this section.

All references should be reviewed to fill the format requirements. Also within the text to unify the format of reference numbers with a space before the parentheses.

6. PLOS authors have the option to publish the peer review history of their article (what does this mean?). If published, this will include your full peer review and any attached files.

**Do you want your identity to be public for this peer review?** For information about this choice, including consent withdrawal, please see our Privacy Policy.

Reviewer #1: No

Reviewer #2: No

---

## [Decision Letter · Decision Letter 1]

5 Oct 2023

PGPH-D-23-01088R1

Feasibility of a social protection linkage program for individuals at-risk for tuberculosis in Uganda

Dear Dr. Shete,

Thank you for submitting your manuscript to PLOS Global Public Health. After careful consideration, we feel that it has merit but does not fully meet PLOS Global Public Health’s publication criteria as it currently stands. Therefore, we invite you to submit a revised version of the manuscript that addresses the points raised during the review process.

We look forward to receiving your revised manuscript.

Kind regards,

Lei Gao

Academic Editor

Journal Requirements:

Additional Editor Comments (if provided):

The manuscript improved largely after the first-round revision. Please submit a revision version according to the minor comments from reviewers.

Reviewers' comments:

Reviewer's Responses to Questions

**Comments to the Author**

1. If the authors have adequately addressed your comments raised in a previous round of review and you feel that this manuscript is now acceptable for publication, you may indicate that here to bypass the “Comments to the Author” section, enter your conflict of interest statement in the “Confidential to Editor” section, and submit your "Accept" recommendation.

Reviewer #1: (No Response)

Reviewer #2: All comments have been addressed

2. Does this manuscript meet PLOS Global Public Health’s publication criteria? Is the manuscript technically sound, and do the data support the conclusions? The manuscript must describe methodologically and ethically rigorous research with conclusions that are appropriately drawn based on the data presented.

Reviewer #1: Yes

Reviewer #2: Partly

3. Has the statistical analysis been performed appropriately and rigorously?

Reviewer #1: N/A

Reviewer #2: No

4. Have the authors made all data underlying the findings in their manuscript fully available (please refer to the Data Availability Statement at the start of the manuscript PDF file)?

Reviewer #1: Yes

Reviewer #2: Yes

5. Is the manuscript presented in an intelligible fashion and written in standard English?

Reviewer #1: No

Reviewer #2: Yes

6. Review Comments to the Author

Reviewer #1: Thank you for addressing almost all of the comments raised on the first draft, however there are a few corrections that should be addressed:

Line 239-240: The word "ineligible" is missing in this sentence.

Line 239-243: Suggest presenting proportions with numerator and denominator e.g. [134/855 (16%)] instead of n=134. This is recommended throughout where proportions are presented in text.

Table 1: Suggest labelling category as "Micro-bacteriologically confirmed TB" instead of "Diagnosed with..."

Table 1: Suggest removing Treated category and presenting Treatment outcome only as: Not treated, Completed, Cured, Transferred out, Lost to follow up, Deceased. Please also include a description of the difference between completed and cured?

Line 319: Suggest rephrasing to .."less than 20% of the 655 who were eligible..." and remove (n=655).

Lline 378-380: Suggest remove or rephrase this statement so that it doesn't sound like new information being presented in the discussion. Perhaps consider rephrasing as a recommendation for future research for the authors?

Reviewer #2: Thank you for your responses, most of the comments were addressed. However, there are still some aspects that could improve the work.

Regarding statistical analysis, it is not necessary to generalize results, but the relationship between characteristics can help, for example, to understand the relationship between sex and the result of a specific sample.

Furthermore, as the authors mentioned that this is a study to evaluate the results of implementation and enrollment, it is suggested to define the outcome as feasibility, which is a composite outcome, with indicators as described in line 198.

Tables were improved, it is suggested in tables 1 and 2, for the last column title only to include the most frequent unit (example: n(%)) used in the table, and for those variables with a different unit of measurement to indicate it in the row next to the variable name.

It is not clear how acceptability was assessed, which is mentioned several times.

It is not recommended to refer to tables from results in the discussion.

Although the text has been improved, there are still some spaces missing before reference parenthesis.

Abstract results can be revised to make them clearer, for example how many participants were interviewed.

7. PLOS authors have the option to publish the peer review history of their article (what does this mean?). If published, this will include your full peer review and any attached files.

**Do you want your identity to be public for this peer review?** For information about this choice, including consent withdrawal, please see our Privacy Policy.

Reviewer #1: No

Reviewer #2: No

---

## [Editor Report · Decision Letter 2]

7 Nov 2023

Feasibility of a social protection linkage program for individuals at-risk for tuberculosis in Uganda

PGPH-D-23-01088R2

Dear Dr Shete,

We are pleased to inform you that your manuscript 'Feasibility of a social protection linkage program for individuals at-risk for tuberculosis in Uganda' has been provisionally accepted for publication in PLOS Global Public Health.

Best regards,

Lei Gao

Academic Editor